# Pulmonary Sarcoidosis: Diagnosis and Differential Diagnosis

**DOI:** 10.3390/diagnostics11091558

**Published:** 2021-08-28

**Authors:** Nicol Bernardinello, Simone Petrarulo, Elisabetta Balestro, Elisabetta Cocconcelli, Marcel Veltkamp, Paolo Spagnolo

**Affiliations:** 1Respiratory Disease Unit, Department of Cardiac, Thoracic, Vascular Sciences and Public Health, University of Padova, Via Giustiniani 2, 35128 Padova, Italy; nicol.bernardinello@unipd.it (N.B.); simone.petrarulo91@gmail.com (S.P.); elisabetta_balestro@hotmail.com (E.B.); ecocconcelli@icloud.com (E.C.); 2Department of Pulmonology, ILD Center of Excellence, St. Antonius Hospital, 3430 EM Nieuwegein, The Netherlands; m.veltkamp@antoniusziekenhuis.nl

**Keywords:** sarcoidosis, diagnosis, histology, imaging, biomarkers, differential diagnosis

## Abstract

Sarcoidosis is a multisystem disorder of unknown origin and poorly understood pathogenesis that predominantly affects lungs and intrathoracic lymph nodes and is characterized by the presence of noncaseating granulomatous inflammation in involved organs. The disease is highly heterogeneous and can mimic a plethora of other disorders, making diagnosis a challenge even for experienced physicians. The evolution and severity of sarcoidosis are highly variable: many patients are asymptomatic and their disease course is generally benign with spontaneous resolution. However, up to one-third of patients develop chronic or progressive disease mainly due to pulmonary or cardiovascular complications that require long-term therapy. The diagnosis of sarcoidosis requires histopathological evidence of noncaseating granulomatous inflammation in one or more organs coupled with compatible clinical and radiological features and the exclusion of other causes of granulomatous inflammation; however, in the presence of typical disease manifestations such as Löfgren’s syndrome, Heerfordt’s syndrome, lupus pernio and asymptomatic bilateral and symmetrical hilar lymphadenopathy, the diagnosis can be established with high level of certainty on clinical grounds alone. This review critically examines the diagnostic approach to sarcoidosis and emphasizes the importance of a careful exclusion of alternative diagnoses.

## 1. Introduction

Sarcoidosis is a chronic systemic disorder of unknown origin characterized histologically by the presence of noncaseating granulomas in affected organs. Although the disease pathogenesis remains poorly understood, sarcoidosis is believed to result from a complex interaction between host/genetic and environmental/infectious factors leading to an aberrant immune response [1]. The lungs and intrathoracic lymph nodes are almost invariably involved, but sarcoidosis can affect any organ system [2,3]. Because no specific diagnostic test or *gold standard* exists, the diagnosis of sarcoidosis is most likely to be correct if compatible clinical and radiological features are supported by the presence of noncaseating granulomas in affected organs and after exclusion of other causes of granulomatous inflammation. Indeed, sarcoidosis can mimic a large number of conditions, including, among others, infection, vasculitis, drug reaction and malignancy. However, a diagnosis of sarcoidosis can be established on clinical ground alone in the following scenarios: asymptomatic patients with bilateral hilar lymphadenopathy (BHL; radiological stage I), Löfgren’s syndrome, Heerfordt’s syndrome and *lupus pernio* (purplish papules or plaques that usually involve the nose, cheeks, lips, ears and eyelids). In such cases, a histological confirmation of the diagnosis is not required, but close clinical follow-up is advised [4]. Sarcoidosis is generally benign and the majority of patients do not require treatment, with spontaneous resolution occurring in approximately 50% of cases [5]. However, approximately one-third of patients develop chronic or progressive disease [6], and about 5% will eventually die, mainly from pulmonary or cardiovascular complications [7,8,9].

This review aims to summarise recent advances in the diagnosis of sarcoidosis, with emphasis on the role of imaging and the importance of a careful exclusion of alternative diagnoses.

## 2. Epidemiology and Definition

Sarcoidosis occurs worldwide and most commonly affects young and middle-aged individuals of all races and genders. The precise annual incidence and prevalence of the disease are unknown, mainly due to the variable methods of cases ascertainment, lack of consistent case definition and heterogeneous disease manifestations. In addition, in a minority of cases, sarcoidosis remains undiagnosed unless radiographic screening is undertaken [10]. With these limitations, it is well established that the incidence of sarcoidosis varies widely across geographical areas, age groups and races, being more frequent in Scandanavian countries and in African Americans, who also tend to be affected with more severe disease than patients of other ethnicities [2,11]. On the contrary, the incidence of sarcoidosis is lower in Hispanic and Asian populations [12].

Overall, the average age of onset in most studies is between 47 and 51 years, with the peak ages of onset closer to 30–55 years [12]. Sarcoidosis is rare in the pediatric population, whereas approximately 30% of cases occur in elderly patients [13].

## 3. Clinical Features

Sarcoidosis has numerous clinical manifestations, but respiratory tract involvement occurs at some point in nearly all patients. Dry cough, dyspnea and chest discomfort are the most common symptoms and are generally more prominent in patients with significant endobronchial or parenchymal involvement [14]. The skin, eyes, liver, spleen and peripheral lymph nodes are the next most common disease sites, with the frequency of involvement ranging from 10% to 30% [15]. Cardiac and neurological disease are uncommon but potentially lethal complications [16,17]. Manifestations such as small fiber neuropathy and fatigue are not organ-specific and have been reported in as many as 86% and more than 90% of patients, respectively [18,19].

Some forms of sarcoidosis are characterized by specific constellation of manifestations: Löfgren’s syndrome, which is generally acute, benign and self-limiting, is characterized by fever, BHL, erythema nodosum, and periarthritis [20], whereas Heerfordt’s syndrome is characterized by parotid or salivary glands enlargement, fever, facial nerve paralysis and anterior uveitis [2,21].

## 4. Pulmonary Function Tests

Pulmonary function tests (PFTs) are useful to assess the severity of respiratory involvement and to monitor the disease course. Different from other interstitial lung disease such as idiopathic pulmonary fibrosis, pulmonary function tests in sarcoidosis do not provide an accurate estimate of the extent of parenchymal disease nor can they predict response to treatment; in fact, some patients could present an extended parenchymal radiological impairment with normal or minimally altered PFT.

The majority of patients with sarcoidosis have a normal lung function [22]. In patients with abnormal PFTs both restrictive as well as obstructive patterns can be found, but the most noted abnormality is a reduced diffusing capacity of the lung for carbon monoxide (DL_CO_) [22,23]. Furthermore, significant differences in PFT parameters among patients with different radiographic stages of sarcoidosis have been shown [22].

## 5. Imaging

### 5.1. Chest Radiography

Pulmonary imaging has a key role in the diagnosis of sarcoidosis, and all patients evaluated for suspected sarcoidosis should have a chest X-ray. This is abnormal in more than 90% of cases, with bilateral hilar lymphadenopathy and parenchymal changes being observed in 50 to 85% and 25 to 60% of cases, respectively [24]. In the 1960s, Guy Scadding developed a “staging system” for pulmonary sarcoidosis using chest radiography: stage I is defined by the presence of BHL; stage II consists of BHL and parenchymal infiltrates; stage III is characterized by parenchymal abnormalities without BHL; stage IV consists of upper-lobe predominant fibrotic changes with volume loss [25]. The frequencies of the different stages at presentation are 50% for stage I, 25–30% for stage II, 10–12% for stage III and 5% for stage IV [26].

Radiographic resolution occurs in three-quarters of patients with stage I and two-thirds of patients with stage II. Patients with higher radiographic stages tend to have more pulmonary symptoms, greater functional impairment and lower likelihood of remission. However, due to the significant overlap between stages, the Scadding system alone cannot be used to predict outcome in individual patients [27]. In addition, there is not necessarily a progression to a higher stage, and, with the exception of stage IV, radiographic resolution may occur with any stage.

### 5.2. Computed Tomography (CT Scan)

Chest high resolution CT (HRCT) is more sensitive than chest X-ray and provides a more precise assessment of hilar, mediastinal and parenchymal abnormalities (Figure 1). A typical HRCT feature in sarcoidosis is the presence of well-defined micronodules scattered along the broncho-vascular bundle, veins, fissures and pleura in a characteristic lymphatic distribution [28]. The micronodules can be confluent, leading to mass-like conglomerations and distortion of the lung parenchyma [29]. Occasionally, conglomerate masses are surrounded by a multitude of micronodules, hence the term “galaxy sign”, which is highly suggestive of pulmonary sarcoidosis. All these abnormalities typically display a mid-to-upper zone predominance.

Additional HRCT findings include ground glass opacities and septal and nonseptal lines [30]. Up to 20% of cases develop pulmonary fibrosis, which manifests on HRCT as bronchial distortion (40% of cases), honeycombing (26% of cases) and linear scarring (14% of cases) [31,32]. Complications of fibrotic pulmonary sarcoidosis include pulmonary hypertension and mycetoma, with hemoptysis representing a rare but potentially life-threatening manifestation [33].

### 5.3. Positron Emission Tomography (FDG PET/CT)

Fluorine 18 fluorodeoxyglucose (FDG) PET/CT is a useful tool for assessing sarcoidosis activity, with a sensitivity of 89 to 100% [34]. In addition, FDG PET/CT may be useful for identifying occult disease (i.e., cardiac or osseous sarcoidosis), or the most suitable site to biopsy and for assessing disease extent and response to treatment [34,35].

In a small study by Mostard et al., more than 90% of patients with fibrotic lesions (Scadding stage IV) still had active inflammatory disease as measured by FDG-PET [36]. Additional studies are needed to further confirm the utility of PET imaging in the diagnosis and assessment of treatment response in sarcoidosis [37].

## 6. Confirmation of the Diagnosis

### 6.1. Fiberoptic Bronchoscopy

Because of the high prevalence of pulmonary involvement, bronchoscopy, with its ancillary sampling techniques, has the highest diagnostic yield in sarcoidosis, unless more easily accessible biopsy sites, such as skin or superficial lymph nodes, are available [38]. Bronchoscopy may reveal a cobblestone appearance of the mucosa—the typical feature of endobronchial granulomatous inflammation. In such cases, the diagnostic yield of endobronchial biopsy is >70%, but it is only about 30% if the airway mucosa appears normal (Table 1) [39].

### 6.2. Bronchoalveolar Lavage

Although the main role of BAL in sarcoidosis is to exclude alternative diagnoses, such as, among others, infection and malignancy, it may also provide supportive evidence for a diagnosis of sarcoidosis, provided BAL data are interpreted in the context of clinical and radiological features. An increased cellular count with a lymphocytosis > 25% and an elevated CD4/CD8 ratio are common findings, though they are neither sensitive nor specific for a diagnosis of sarcoidosis. Indeed, the BAL cell count may be normal in approximately 15% of newly diagnosed cases [40]. However, a CD4/CD8 ratio > 3.5 has a specificity of 93–96%, although with a sensitivity of 53 to 59% and a CD4/CD8 ratio > 10 has a >99% specificity for a diagnosis of sarcoidosis [41]. Additional features suggestive of a diagnosis of sarcoidosis include a lymphocytosis > 15%, CD4/CD8 ratio > 3.5, CD103 ratio of <0.2 and a transbronchial biopsy demonstrating noncaseating granulomas, whereas a CD4/CD8 ratio < 1 and elevated neutrophil and eosinophil counts make the diagnosis of sarcoidosis unlikely [41,42,43]. Patients with severe and progressive disease may show increased BAL neutrophils, which portend a poor response to immunosuppressive therapy [44].

### 6.3. EBUS-TBNA

Mediastinal lymphadenopathy is the most common manifestation of sarcoidosis across all ethnic groups [29], and different techniques can be used to sample the mediastinum, including endobronchial ultrasound with real-time guided transbronchial needle aspiration [EBUS-TBNA], transesophageal endoscopic ultrasound-guided fine-needle aspiration [EUS-TBNA] and endoscopic ultrasound bronchoscope guided fine-needle aspiration [EUS-b-FNA]. In particular, EBUS-TBNA can safely sample virtually any nodes in contact with the large airways and, when coupled with cytologic evaluation, has a consistently high success rate in diagnosing sarcoidosis [45,46], particularly in stage I and stage II disease [47]. Indeed, its diagnostic yield in stage I disease is 84% (95% CI, 74–92%) compared with 77% (95% CI, 64–86%) in stage II. In patients with hilar and/or mediastinal adenopathy, EUS-TBNA performs better than EBUS-TBNA, with a diagnostic yield of 88% (95% CI, 80–93%) vs. 66% (95% CI, 53–77%) [48]. However, EBUS-TBNA gives better access to the lymph nodes commonly involved in sarcoidosis than EUS–FNA [37,49]. Importantly, endosonography may help differentiating sarcoidosis from an adenopathy of a different nature, such as tuberculosis, based on sonographic features (i.e., the presence of heterogeneous echotexture in B-mode or necrosis in intrathoracic lymph nodes in a patient with positive tuberculin skin test (TST), especially in countries with high tuberculosis burden, may support a diagnosis of tuberculosis rather than sarcoidosis) [50]. In patients with suspected sarcoidosis, it is recommended to sample more than one nodal station. Indeed, the sampling of two lymph node stations versus one was significantly associated with the likelihood of a positive needle aspirate or biopsy sample [51]; the best success rates were obtained with aspirates or biopsies performed in the right paratracheal and subcarinal areas.

### 6.4. Transbronchial Lung Biopsy (TBLB)

Transbronchial Lung Biopsy (TBLB) has a relatively high diagnostic yield (50 to 75%) in patients with suspected sarcoidosis based on bilateral hilar adenopathy or compatible lung parenchymal abnormalities on HRCT. Conversely, in patients with stage I disease, the diagnostic accuracy of TBLB is suboptimal (12–66%), at least when four to five biopsies per patient are obtained [52]. When coupled with EBUS-TBNA, the diagnostic yield of TBLB (versus TBLB alone) increases from 82% to about 93% [53,54]. The parenchymal areas to sample should be chosen based on CT findings, avoiding areas with significant architectural distortion and, in cases with mild radiological abnormalities, the right lower lobe segments [54]. TBLB is generally a safe procedure, although bleeding and pneumothorax are uncommon but well-established complications [55]. The diagnostic yield of TBLB positively correlates with the number and size of samples obtained; the highest success rates are achieved when 8–10 biopsies are obtained [56], but such a high number of biopsies is seldom obtained in clinical practice. EBUS-TBNA is the preferred technique in the diagnostic work-up of patients with suspected stage I sarcoidosis. However, in order to obtain the highest success rate in the detection of granulomas, EBUS-TBNA needs to be combined with TBLB [53]. Recent data suggest that transbronchial lung cryobiopsy (TBLC) may be useful in cases without significant lymphadenopathy and parenchymal abnormalities and in atypical cases when other sampling techniques are inconclusive, thus avoiding surgical lung biopsy [57]. However, more data are needed to clarify the role of TBLC in the diagnostic work-up of patients with suspected sarcoidosis.

### 6.5. Mediastinoscopy

If less invasive tests are not diagnostic or inconclusive, the next steps to sample the mediastinal lymph nodes are mediastinoscopy [58], and, in highly selected cases, thoracotomy (open or video-assisted thoracoscopy). However, surgical procedures are invasive, expensive and associated with longer hospital stays and significant morbidity, with prolonged air leakage being the most common postoperative complication. The benefits of lung biopsy to secure a histological diagnosis of sarcoidosis remain controversial, and many clinicians are reluctant to suggest surgical procedures to their patients due the non-negligible operative risks. In cases that remain undiagnosed following less invasive procedures and in the presence of mediastinal lymphadenopathy, mediastinoscopy should be the preferred sampling modality [59].

### 6.6. Serological Biomarkers

As with other interstitial lung diseases, diagnostic, prognostic and theragnostic biomarkers are also lacking in sarcoidosis. Angiotensin converting enzyme (ACE) and the soluble interleukin-2 receptor (sIL-2R) are the most widely used biomarkers in sarcoidosis, though with suboptimal sensitivity and specificity. ACE is produced by the epithelioid cells within the sarcoid granuloma, and high levels of serum ACE are believed to reflect the burden of granulomatous inflammation [63,64]. However, elevated ACE levels may also be found in other granulomatous disorders such as chronic beryllium disease (CBD) or leprosy, as well as in liver disease, lymphoma, diabetes and hyperthyroidism [65]. In addition, ACE inhibitors, which are widely used antihypertensive drugs, reduce serum ACE level, thus leading to false negative values [66]. Furthermore, an insertion/deletion polymorphism within the ACE gene is known to affect plasma ACE levels [67]. Therefore, the value of ACE levels as diagnostic or prognostic tool remains a matter of debate.

sIL-2R is released by activated mononuclear cells and sIL2R levels correlate with disease activity, particularly in patients with extrapulmonary disease. In addition, sIL2R is more sensitive than serum ACE and lysozyme levels for a diagnosis of sarcoidosis [68], although sIL-2R levels can also be elevated in hematologic malignancy, autoimmune disorders and idiopathic pulmonary fibrosis [69]. A more recent study in patients with suspected sarcoidosis demonstrated a sensitivity of 88% and a specificity of 85% [70]. This indicates that sIL-2R can be a useful tool in the diagnosis of sarcoidosis when combined with imaging and clinical features.

To date, neither ACE nor sIL2R measurement are recommended as routine tests in the diagnostic work-up, initial assessment or follow up of patients with sarcoidosis [4].

## 7. Differential Diagnosis

Due to the lack of a diagnostic gold standard and the absence of a specific aetiology, the diagnosis of sarcoidosis remains one of exclusion [37]. In addition, granulomatous inflammation is the histologic hallmark of a number of conditions, including bacterial, mycobacterial and fungal infection, and occupational lung diseases such as chronic beryllium disease and silicosis. Occasionally, granulomatous inflammation may result from an immunological response against neoplastic antigens or drugs (Table 2).

### 7.1. Tuberculosis and Other Infectious Diseases

Infection, particularly tuberculosis (TB), is always in the differential diagnosis of patients suspected of having sarcoidosis. In countries where TB is endemic, differentiating between the two may be challenging, although a number of differences can also be appreciated. Unlike sarcoidosis, TB is characterized by the presence of a necrotizing granuloma in affected organs and cultures positive for Mycobacterium tuberculosis. It is important to state, however, that both features may be absent. In such cases, a positive Interferon-gamma release assay (IGRA) or tuberculin skin test (TST) associated with a positive PCR on clinical specimens generally suggest the correct diagnosis [71]. Rarely (<1% of cases), sarcoidosis may mimic miliary TB, with diffuse micronodules randomly disseminated in both lungs [72]. Of note, a multicentre study by Wang and colleagues revealed that TB patients are at higher risk of developing sarcoidosis compared to non-TB subjects [73]. Other infectious diseases such as histoplasmosis, coccidioidomycosis, brucellosis and nocardiosis can also mimic sarcoidosis. In these cases, where other microbiological causes than TB are suspected, blood tests, bronchoalveolar, hematological and urine cultures could be useful for a correct diagnosis.

### 7.2. Occupational and Environmental Exposure: Chronic Beryllium Disease and Silicosis

Clinical, radiological and histological features of CBD are virtually indistinguishable from those of sarcoidosis [74]. Indeed, CBD is commonly referred to as “sarcoidosis of known cause”. Beryllium, a metal that is used in gears and cogs particularly in the aviation industry, can be extremely harmful if inhaled. The clinical course of CBD is highly variable, with approximately one-third of patients developing a progressive decline in lung function to end-stage lung disease [75]. The diagnosis of CBD requires a combination of confirmed beryllium sensitization (i.e., two or more positive beryllium lymphocyte proliferation assays (BeLPTs), ideally on BAL cells) along with pathological evidence of granulomatous inflammation [76].

Silicosis represents a spectrum of pulmonary diseases (Figure 2) caused by the inhalation of free crystalline silicon dioxide (silica). The diagnosis of silicosis requires a history of silica exposure that is sufficient to cause the disease, compatible radiographic (usually a conventional chest radiograph) abnormalities and the exclusion of other illnesses that can mimic silicosis [74]. Enlarged hilar lymph nodes, a common finding in silicosis, tend to calcify circumferentially in 5 to 10% of cases, producing the so-called eggshell pattern of calcification [77]. The link between silica and sarcoidosis is interesting, since previous retrospective studies demonstrated a higher risk of sarcoidosis in workers with occupational silica exposure [78,79]. In one case report, a confirmed exposure to silica together with immunological sensitization to silica was described in a patient diagnosed with sarcoidosis [80]. Therefore, as highlighted by the recent British Thoracic Society (BTS) clinical guideline on sarcoidosis, it is important to take an accurate exposure and occupational history to exclude conditions such as berylliosis or silicosis [81].

### 7.3. Lymphoma, Cancer and Drug: The Sarcoid-like Reaction 

Non-necrotizing granulomas may be seen in a number of settings, including malignancy, drug toxicity and following medical device implantation [82]. The presence of (isolated) non-necrotizing granulomatous inflammation in patients otherwise not meeting clinical/radiological criteria for sarcoidosis is referred to as a sarcoid-like reaction (SLR). The prevalence of SLRs (Figure 2) is about 4% in patients with solid neoplasm and 14% and 7% in patients with Hodgkin’s disease and non-Hodgkin lymphoma, respectively [83]. Notably, Murthi and colleagues have recently reported that the presence of SLRs is associated with a reduced number of metastases and better survival [84], although the mechanisms underlying the “protective” effect of SLRs are unclear. SLRs are generally observed in lymph nodes near solid tumors or occasionally in more distant sites such as the spleen, liver, and bone marrow. [85] SLRs can also occur following antineoplastic treatment, including Bacillus Calmette-Guérin (BCG) for bladder cancer [86], anti-cytotoxic T-lymphocyte antigen 4 (CTLA4) [87], anti-programmed death-ligand 1 (PD-L 1) [88,89] and nivolumab [90]. In this setting, SLRs are referred to as Drug Induced Sarcoid-Like Reactions (DISR) [91]. Other therapies that can induce SLRs include antiretroviral drugs, adalimumab [92] and infliximab [93], which, paradoxically, are also potential therapies for sarcoidosis. The exact immunopathogenesis of DISR is currently unknown. Finally, SLRs have been reported near silicone breast implants and following joint replacement [94] and body piercing [95].

### 7.4. Common Variable Immunodeficiency

Common variable immunodeficiency (CVID), the most common primary immunodeficiency disorder, is a syndrome characterized by low serum antibody levels, poor antibody response and recurrent bacterial infection [96]. CVID is one of the most common forms of immunodeficiency in adults and is rare in children, where other forms may be more frequent. CVID may manifest as granulomatous disease, which may be difficult to distinguish from sarcoidosis, particularly when granulomatous inflammation is widespread and precedes the onset of the immune deficiency [97]. On CT, CVID is characterized by the presence of ill-defined (either centrilobular or randomly distributed) nodules with a mid-lower lobe predominance, although well-defined nodules with a perilymphatic distribution as well as hilar and mediastinal lymphadenopathy have also been reported [28]. Similar to sarcoidosis, a histological examination of CVID lesions reveals non-necrotizing granulomas, although the presence of areas of organizing pneumonia and follicular bronchiolitis, both uncommon in sarcoidosis, may help in distinguishing the two conditions. Low serum immunoglobulin levels along with a history of recurrent infection strongly favor a diagnosis of CVID over sarcoidosis.

### 7.5. Autoimmune Disorders

Due to the heterogenic presentation, sarcoidosis must be careful to also be distinguished from rheumatologic disorders and vasculitis. At presentation, many patients with sarcoidosis could present symptoms similar to autoimmune diseases such as arthralgia/arthritis, parotitis, weight loss, cutaneous lesions, asthenia and fever [2]. Laboratory tests (antinuclear antibodies (ANA); antibodies against extractable nuclear antigens (ENA); anticitrullinated peptide antibodies (ACPA); rheumatoid factor (FR); and others), radiological imaging and histological studies are necessary for a correct diagnosis [98]. However, in a study conducted by Wu C.H. in 1237 sarcoidosis patients, 17.6% presented an overlap syndrome with another autoimmune or inflammatory diseases [99].

Finally, the usual interstitial pneumonia (UIP) pattern could be present in pulmonary fibrotic sarcoidosis, ILD associated with rheumatoid arthritis and systemic scleroderma [31,100].

## 8. Conclusions

Sarcoidosis is an uncommon disorder of unknown aetiology that can mimic a multitude of other diseases. The diagnosis remains one of exclusion and requires compatible clinical and radiological features together with the evidence of (non-necrotizing) granulomatous inflammation at disease sites, with the exception of some well-defined clinical manifestations such as Löfgren’s syndrome, Heerfordt’s syndromes and lupus pernio. The diagnosis of sarcoidosis may be challenging even for experienced clinicians because multiple other granulomatous disorders can mimic sarcoidosis and need to be excluded. Higher-quality evidence is urgently needed to guide clinical practice with regard to the diagnosis and detection of sarcoidosis.

## Figures and Tables

**Figure 1 diagnostics-11-01558-f001:**
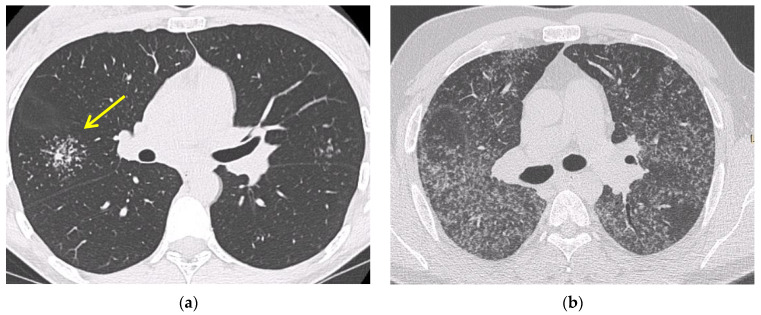
Irregularly marginated nodule surrounded by multiple small nodules (“Galaxy sign”, yellow narrow), this is typical of sarcoidosis (**a**); ground-glass-like increased attenuation resulting from diffuse micronodules randomly distributed (“Miliary sarcoidosis”) (**b**); enlarged and partially calcified (yellow narrows) bilateral hilar lymph nodes (**c**); fibrotic sarcoidosis with cystic changes and traction bronchiectases (yellow narrows) predominantly in the perihilar region and upper lobes. Nodular abnormalities are minimal/absent, but the appearance and the location of the fibrosis are very suggestive of the diagnosis of sarcoidosis (**d**).

**Figure 2 diagnostics-11-01558-f002:**
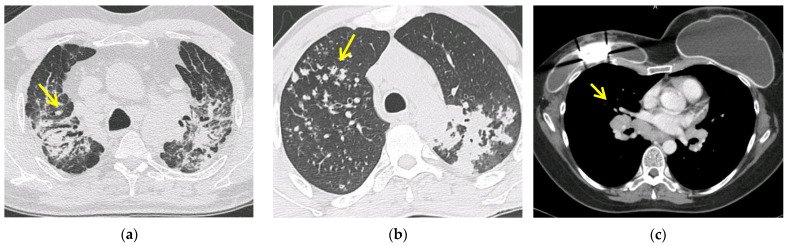
Pulmonary silicosis with predominantly upper lobes fibrosis and traction bronchiectasis (yellow narrow) (**a**); conglomerate masses and pulmonary micronodules (yellow narrow) in patient with tuberculosis (**b**); enlarged bilateral hilar lymph nodes (yellow narrow), a sarcoid-like-reaction in patient with mammal carcinoma (**c**).

**Table 1 diagnostics-11-01558-t001:** Sensitivity and specificity of the main diagnostic procedures in suspicion of pulmonary sarcoidosis.

	Sensitivity	Specificity	Diagnostic Yield	References
EBB	46.2%	85.7%	30–70%	[39]
TBLB	37%	100%	50–75%	[52,60]
EBUS/TBNA	83–93%	100%	77–84%	[48,61]
Mediastinoscopy	100%	100%	82–100%	[58]
BAL (CD4/CD8 ≥ 3.5)	53–59%	93–96%	56%	[41,62]

EBUS/TBNA: endobronchial ultrasound with real-time guided transbronchial needle aspiration; EBB: endobronchial biopsy; TBLB: transbronchial lung biopsy; BAL: bronchoalveolar lavage.

**Table 2 diagnostics-11-01558-t002:** Main differential diagnoses of sarcoidosis.

	Sarcoidosis	Tuberculosis	CBD and Silicosis	Sarcoid-like Reactions (SLRs)
Clinical presentation	Often asymptomatic May be an occasional diagnosis Dry cough, dyspnea Weight loss Fever	Weight loss Cough Purulent sputum Hemoptysis Fever	Dry cough and dyspnea	Often Asymptomatic
Exposure history	Undefined	Recent travel to endemic countries, contact with TB patient	History of work/environment exposure to beryllium or silica	Drugs, malignancy or medical device implantation
Radiological findings or localizations	Bilateral and symmetrical hilar lymphadenopathy Perilymphatic and peribronchovascular nodules Cavitation (rare)	Hilar lymphadenopathy (often asymmetrical) Cavitation (frequent) Randomly distributed nodules	Bilateral hilar lymphadenopathy (CBD); lymph nodes may have an egg-shell appearance (silicosis)	It depends on the underlying cause (i.e., lymph nodes near solid tumors)
Laboratory	Hypercalcemia and hypercalciuria Elevated serum levels of ACE Elevated levels of sIL-2R Peripheral lymphopenia Mantoux test: anergic	Mantoux test: positive IGRA: positive ACE levels may be elevated	Mantoux test: negative ACE levels may be elevated	It depends on the underlying cause ACE may occasionally be elevated Mantoux test: negative
Histopathology	Nonnecrotizing granulomas	Necrotizing granulomas	Nonnecrotizing granulomas Sclerotic nodules Silica particles	Indistinguishable from sarcoid granulomas
Bronchoscopy and BALF	Lymphocytosis CD4+/CD8+ ratio generally > 3.5	Culture positive for mycobacterium tuberculosis	Lymphocytosis Positive BeLPT (CBD)	Variable based on the underlying cause

BALF: bronchoalveolar lavage fluid; ACE: angiotensin converting enzyme; CBD: chronic beryllium disease; BeLPT: beryllium lymphocyte proliferation test; IGRA: Interferon-gamma release assay; sIL-2R: soluble interleukin-2 receptor.

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
