# Peer review of "Pulmonary Sarcoidosis: Diagnosis and Differential Diagnosis"

_diagnostics, 2021, doi:10.3390/diagnostics11091558_

Round 1

Reviewer 1 Report

The paper is well written, without errors in the text, but is not very interesting; the fact is that this paper contributes little that is new to literature about sarcoidosis. There are already a lot of articles about differential diagnosis of sarcoidosis. Moreover, there are few images in the text, it would be better if there were radiographic images of differential diagnosis presented in the text.

Author Response

REV 1

The paper is well written, without errors in the text, but is not very interesting; the fact is that this paper contributes little that is new to literature about sarcoidosis. There are already a lot of articles about differential diagnosis of sarcoidosis. Moreover, there are few images in the text, it would be better if there were radiographic images of differential diagnosis presented in the text.

Comment:

We thank the referee for her/his suggestion about images and we amended the manuscript adding more pictures (Figure 2, page 10) regarding three patients with different diagnoses, in particular affected by silicosis, tuberculosis and sarcoid like reaction.

Differently from an original research paper, the aim of the this manuscript (review article) was to provide an overview of current literature focusing on the topic of differential diagnosis and diagnostic procedures regarding pulmonary sarcoidosis. To this aim we drew the article according to recent guideline (reference n.4).

Reviewer 2 Report

The paper is well written and thorough.  Only a few minor questions/corrections are needed for publication.

Section 2 Epidemiology: Should discuss the regional variances in sarcoidosis with a prevalence in Scandanavian countries and lower incidence in Spain and Japan (Newman et al. NEJM 1997, 336:1224.

Line 63: Change Back American to African American

Line 80: Lofgren's is also associated with a more acute onset

Section 4 Imaging: Can the authors describe the percentages of the different stages at the time of diagnosis?

Line 132: What is the sensitivity of PET in diagnosing sarcoid?  

Author Response

The paper is well written and thorough. Only a few minor questions/corrections are needed for publication.

- Section 2 Epidemiology: Should discuss the regional variances in sarcoidosis with a prevalence in Scandanavian countries and lower incidence in Spain and Japan (Newman et al. NEJM 1997, 336:1224.

- Line 63: Change Back American to African American

- Line 80: Lofgren's is also associated with a more acute onset

- Section 4 Imaging: Can the authors describe the percentages of the different stages at the time of diagnosis?

- Line 132: What is the sensitivity of PET in diagnosing sarcoid?  

Comment:

We thank the referee for her/his suggestion. We then included in the manuscript data regarding the regional variance in sarcoidosis (page 2 lines 61-65), the percentage of the different Scadding stage at presentation (page 3 lines 106-108) and add the sensitivity of PET CT (page 3 lines 133-134).

Reviewer 3 Report

The authors summarize a clinical approach to sarcoidosis and provide guidance on differential diagnoses. As there are many reviews in the field of sarcoidosis, with every additional review one has to try to find a novel or different angle to avoid overlap with other, similarly tuned papers. 

I, therefore, suggest the following changes to the manuscript: 

Title and abstract:

- The title is a little bit misleading, as the paper mainly focuses on "pulmonary sarcoidosis". Would add this in the title.

2. Epidemiology: "In addition, in a large minority...".- do you mean small minority (or just "minority"?)

3. Clinical features: should change "arthralgia" in löfgren's to "periarthritis"; add "fever" to Heerfordt's? Not sure I would add PFTs to clinical features. Also, expand the last sentence a bit on the correlation of PFT with radiological stages. 

4.IMaging: suggest renaming the subheadings: "chest radiography", "computed tomography" "positron emission tomography"

  • authors should add the Scadding stages only apply to CXR, sometimes there is confusion in clinical practice and it is used for CT scans as well.
  • Figure1: add arrows / arrowheads to the CT scans to highlight the changes

5. Confirmation of the diagnosis:

  • here, I suggest that the authors add an algorithm or tabular overview on the use of the different techniques. this would really add extra information and discern from other reviews on the same topic.

6. DIfferential diagnosis

  • add sIL-2R to sarcoid in the table?
  • expand a bit on infectious differential diagnoses other than TB (how to distinguish, tests, microbiology stains?)
  • I like the fact that CVID is mentioned. Add: it is one of the most common forms of immunodeficiency in adults (selective IgA def more common), in children, other forms may be more common. What about bronchiectasis in CVID?

The authors should mention the recently published ATS statement: https://www.atsjournals.org/doi/10.1164/rccm.202002-0251ST (haven't seen it referenced). 

Minor typos: 

Abstract: "... and requires long-term therapy."; "...clinical grounds alone."

Intro p1 line 32: Although the disease pathogenesis... line39: noncaseating "granulomas"?

p5 line199: high tuberculosis burden 

p6 line 241: high levels, line 246: false negative 

Author Response

REV 3:

The authors summarize a clinical approach to sarcoidosis and provide guidance on differential diagnoses. As there are many reviews in the field of sarcoidosis, with every additional review one has to try to find a novel or different angle to avoid overlap with other, similarly tuned papers. I, therefore, suggest the following changes to the manuscript:

1.Title and abstract:

- The title is a little bit misleading, as the paper mainly focuses on "pulmonary sarcoidosis". Would add this in the title.

Comment:

  1. The referee is correct, we then changed the title of the manuscript in “pulmonary sarcoidosis”

  1. Epidemiology:

- "In addition, in a large minority...".- do you mean small minority (or just "minority"?)

Comment:

  1. We meant “minority” and changed it in the manuscript

  1. Clinical features:

- should change "arthralgia" in löfgren's to "periarthritis"; add "fever" to Heerfordt's?

- Not sure I would add PFTs to clinical features. Also, expand the last sentence a bit on the correlation of PFT with radiological stages.

Comment:

  1. We expanded the last sentences on PFTs and radiological stages (page 2 lines 86-90) and amended the article with other suggestions.

4.IMaging:

- suggest renaming the subheadings: "chest radiography", "computed tomography" "positron emission tomography"

  • authors should add the Scadding stages only apply to CXR, sometimes there is confusion in clinical practice and it is used for CT scans as well.
  • Figure1: add arrows / arrowheads to the CT scans to highlight the changes

Comment:

  1. We added the scadding stages (page 3; lines 102-103) and arrows to CT scan (Figure 1,2)

  1. Confirmation of the diagnosis:
  • here, I suggest that the authors add an algorithm or tabular overview on the use of the different techniques. this would really add extra information and discern from other reviews on the same topic.

Comment:

  1. To avoid overlap with previous consensus statement/guideline we decided to add a table regarding “Sensitivity and specificity of the main diagnostic procedures in suspicion of pulmonary sarcoidosis” (page 6)
  2. DIfferential diagnosis
  • add sIL-2R to sarcoid in the table?
  • expand a bit on infectious differential diagnoses other than TB (how to distinguish, tests, microbiology stains?)
  • I like the fact that CVID is mentioned. Add: it is one of the most common forms of immunodeficiency in adults (selective IgA def more common), in children, other forms may be more common. What about bronchiectasis in CVID?
  • The authors should mention the recently published ATS statement: https://www.atsjournals.org/doi/10.1164/rccm.202002-0251ST (haven't seen it referenced).

Comment:

  1. We very appreciated your comment and we amended the manuscript accordingly. The recent published ATS statement was already mentioned (see reference n 4).

Minor typos:

  • Abstract: "... and requires long-term therapy."; "...clinical grounds alone."

  • Intro p1 line 32: Although the disease pathogenesis... line39: noncaseating "granulomas"?

  • p5 line199: high tuberculosis burden

  • p6 line 241: high levels, line 246: false negative

Comment:

We apologize for typos. We rectified them in the manuscript.

Reviewer 4 Report

This is a traditional review, like a chapter in a medical book for students, listing all basic items on sarcoidosis, focus on pulmonary part. 
Major comments 
1. In"Imaging", "Confirmation of the diagnosis", "Differential diagnosis" all these parts are focus on the lung, I would like to suggest to change the title on pulmanory sarcoidosis. 
2.  Differential diagnosis part, it might of certain importance to add some autoimmunal diseases which might affect the lung, and to be distinguished from sarcoidosis.

Minor comments 

  1. A typo in line 63. Back Americans.
  2. It would be better if the authors can add some information on histological studies, as so far the diagnosis of sarcoidosis is based on clinicopathologic findings.

Author Response

REV 4:

This is a traditional review, like a chapter in a medical book for students, listing all basic items on sarcoidosis, focus on pulmonary part.

Major comments

  1. In"Imaging", "Confirmation of the diagnosis", "Differential diagnosis" all these parts are focus on the lung, I would like to suggest to change the title on pulmanory sarcoidosis.
  2. Differential diagnosis part, it might of certain importance to add some autoimmunal diseases which might affect the lung, and to be distinguished from sarcoidosis.

Comment:

We thank the referee for her/his suggestion. We changed the title of the manuscript and added a brief paragraph regarding autoimmune diseases and differential diagnosis with pulmonary sarcoidosis (page 9 line 349-361).

Minor comments

  • A typo in line 63. Back Americans.
  • It would be better if the authors can add some information on histological studies, as so far the diagnosis of sarcoidosis is based on clinicopathologic findings.

Comment:

We agree with the referee about the comment on the importance of histological studies and their contribution on the diagnosis of sarcoidosis. However our manuscript was focused on diagnostic work up from the pulmonologist point of view and aimed to describe procedures and differential diagnoses. We didn’t approach histological characteristics because this not represents the scope of the study. If you consider this aspect essential in this context, we will then add some information regarding this topic.